# Progress on Denosumab Use in Giant Cell Tumor of Bone: Dose and Duration of Therapy

**DOI:** 10.3390/cancers14235758

**Published:** 2022-11-23

**Authors:** Feifan Xiang, Huipan Liu, Jia Deng, Wenzhe Ma, Yue Chen

**Affiliations:** 1State Key Laboratory of Quality Research in Chinese Medicine, Macau University of Science and Technology, Macau 999078, China; 2Department of Orthopedic, Affiliated Hospital of Southwest Medical University, Luzhou 646000, China; 3Department of Nuclear Medicine, Affiliated Hospital of Southwest Medical University, Luzhou 646000, China; 4Nuclear Medicine and Molecular Imaging Key Laboratory of Sichuan Province, Luzhou 646000, China; 5Institute of Nuclear Medicine, Southwest Medical University, Luzhou 646000, China

**Keywords:** giant cell tumor of bone, denosumab, dose, duration of therapy

## Abstract

**Simple Summary:**

Giant cell tumor of bone (GCTB) is an aggressive non-cancerous bone tumor. Surgery remains the main treatment for GCTB. Denosumab has been approved for the treatment of advanced or inoperable GCTB. The short-term efficacy and safety of denosumab in inoperable patients have been demonstrated. Lengthier therapies (high cumulative doses) or pre-operative adjuvant therapy may be associated with severe complications and high local recurrence rates. The impact of differential doses and lengths of treatment on the efficacy of denosumab in GCTB treatment, the incidence of complications, and recurrence rates have gained research traction. Short-term administration helps attain satisfactory local control and functionality. The efficacy and safety of denosumab against GCTB, its impact on imaging assessment, related complications, and recurrence of GCTB were previously reviewed. This paper reviews the progress in studies evaluating the impact of the dose and duration of denosumab therapy for GCTB.

**Abstract:**

Giant cell tumor of bone (GCTB) is an aggressive non-cancerous bone tumor associated with risks of sarcoma and metastasis. Once malignancy occurs, the prognosis is generally poor. Surgery remains the main treatment for GCTB. Multidisciplinary management is a feasible option for patients wherein surgical resection is not an option or for those with serious surgery-related complications. Denosumab is an anti-nuclear factor kappa B ligand approved for the treatment of postmenopausal women with osteoporosis, bone metastases, and advanced or inoperable GCTB. However, the guidelines for treating GCTB are unclear; its short-term efficacy and safety in inoperable patients have been demonstrated. Lengthier therapies (high cumulative doses) or pre-operative adjuvant therapy may be associated with severe complications and high local recurrence rates. Short-term administration helps attain satisfactory local control and functionality. As a result, lately, the impact of different doses and lengths of treatment on the efficacy of denosumab in GCTB treatment, the incidence of complications, and recurrence rates have gained attention. The efficacy and safety of denosumab against GCTB, its impact on imaging assessment, related complications, and recurrence of GCTB were previously reviewed. For further research direction, this paper reviews the progress of studies evaluating the impact of the dose and duration of denosumab therapy for GCTB.

## 1. Introduction

Giant cell tumor of bone (GCTB), a rare primary osteogenic tumor of bone, accounts for approximately 5% of all primary bone tumors [1]. In 2020, the World Health Organization defined GCTB as an intermediate malignant tumor [2]. It usually occurs in the epiphysis of the long bones of the extremities, particularly in the distal femur and proximal tibia. Surgical treatment remains the main therapeutic approach, including either intralesional curettage and cementation/bloc grafting or en bloc resection and reconstruction. Postoperative local recurrence (LR) is the main downside of curettage. However, it has lower overall complication rates than reconstruction after overall resection [3]. The treatment of GCTB in central sites, including the spine and sacrum, remains challenging because it is frequently inoperable and, when operable, severe surgical complications may occur [4]. Additionally, in extremely rare cases, GCTB patients may undergo a sarcomatous transformation and lung metastasis [5,6,7].

The pathophysiology of GCTB has been elucidated, with nuclear factor kappa B ligand (RANKL) being identified as a key player [8]. Furthermore, a driver gene mutation in histone H3.3 is characteristic of GCTB [9]. Denosumab is an anti-RANKL monoclonal antibody that inhibits osteolysis, proliferation, and activity of osteoclasts, with various uses, including the treatment of osteoporosis in postmenopausal women at high risk of fracture, and bone metastases associated with solid tumors [10,11,12]. Additionally, the Food and Drug Administration (FDA) has approved denosumab for the treatment of locally advanced or metastatic GCTB, resulting in a change of the treatment mode from a local to a multidisciplinary approach [13,14]. Denosumab can effectively reduce the number of giant tumor cells in GCTB and induce changes in the histological cell diversity, which may lead to confusion in diagnosis [15,16]. Tumor cells with H3F3A mutations survive after denosumab treatment, thus facilitating its diagnosis [17]. H3.3 G34W positive staining is a specific and sensitive method for detecting H3F3A mutant GCTB [18,19]. Histological changes after treatment with denosumab appear to be related to the length of treatment [20]. The study of the dose and course of denosumab has attracted the attention of researchers.

Numerous basic and clinical studies have shown that denosumab is clinically beneficial for the treatment of advanced or metastatic GCTB [21,22]. However, various controversies regarding its use as a pre-operative adjuvant therapy prevail. Pre-operative adjuvant therapy with denosumab (PATD) may increase the risk of LR in GCTB [23,24,25]. Conversely, a short pre-operative course of denosumab (≤3 doses) may also achieve satisfactory local control and functionality in GCTB [26]. PATD can reduce blood supply and should not be used for more than 3 months [27]. These findings have attracted the attention of researchers, leading to various studies on testing different doses and durations of therapy with denosumab against GCTB.

Several studies have reviewed the safety and efficacy of denosumab therapy in GCTB [28], the induced changes in imaging assessment [29], and the associated rate of LR [30,31]. Therefore, this article reviews the latest pharmacotherapeutic schemes containing denosumab used against GCTB, including the inoperable advanced forms, and as pre-operative adjuvant therapy. Our purpose is to provide a basis for subsequent research and rational prescription in clinical practice.

## 2. The Influence of Dose and Duration of Denosumab Therapy on GCTB Cell Histology, Molecular Parameters, and Diagnosis

Treatment of GCTB with denosumab can significantly reduce or eliminate RANK-positive tumor giant cells but only exerts an inhibitory, and not anti-apoptotic effect, on tumor stromal cells, a histological different cell type in GCTB [32,33,34]. Immunohistochemical testing (IHC) revealed that denosumab treatment resulted in osteoblast differentiation and bone formation. Fibrous bone tissue replaced tumors or osteoclast-like giant cells [35]. After denosumab treatment, tumors only comprising stromal cells still exist in the new bone. These cells continue to proliferate and cause bone destruction, which may be related to LR [36,37]. Expression of the RANKL signaling pathway was detected by IHC in various primary or metastatic bone tumors. Among all the tumors studied, GCTB exhibited the highest RANKL expression and RANKL/OPG ratio. However, the level of mRNA expression of RANKL was also high in aneurysmal bone cysts, fibrous dysplasia, osteosarcoma, chondrosarcoma, and chondroma [38,39]. Denosumab treatment does not regulate the levels of mRNA and protein expression for RANKL or osteoprotegerin (OPG) [40]. Since the post-treatment tumor bears a slight resemblance to the pre-treatment tumor and is easily confused with other bone tumors, it is vital to pay special attention to diagnosing GCTB after treatment with denosumab [41].

Histologically, GCTB is generally thought to consist of three main types of cells: mononuclear tumor-like cells with an osteoblast precursor phenotype, mononuclear histiocytes, and osteoclast-like multinucleated giant cells [42]. GCTB comprises a large number of giant cells, 92% of which harbor the H3F3A mutation [43]. Although denosumab is effective in reducing the number of tumor-like giant cells, H3F3A-mutated tumor cells survive denosumab treatment, and in response, undergo significant histological changes [17]. H3.3 G34W/R/V mutation-specific antibody is an effective surrogate marker for detecting the H3F3A genotype, and thus, a specific and sensitive marker for detecting H3F3A mutations in GCTB [42,44]. The expression of H3.3 G34W mutant protein in GCTB after denosumab therapy suggests that tumor stromal cells may play a role in the formation of a new bone. Therefore, the detection of H3F3A by IHC or sequencing is extremely helpful for the diagnosis of bone tumors lacking giant cells [45,46,47]. Some scholars have compared the sensitivity and specificity of sequencing and IHC for detecting the H3F3A gene for GCTB diagnosis. The sensitivity and specificity of the sequencing method were 100%. Thus, it can be used as the gold standard for the diagnosis of GCTB, and IHC as a census method [48]. H3.3-G34W was knocked down in tumor stromal cells of GCTB and found to affect tumorigenesis in vitro and in vivo. Therefore, H3.3-G34W screening is a promising target for the treatment of GCTB in addition to serving as a diagnostic tool [49].

Wojcik, J. et al. [20] reported that nine GCTB patients treated with denosumab had a significant reduction in giant cells following pathological examination after biopsy or surgery. However, some developed de novo high-grade osteosarcoma, although this sarcoma shows an invasive growth pattern. The duration of their treatment ranged from 2 to 55 months. The histology of GCTB after denosumab treatment is variable and appears to be related to the length of the treatment [20]. Existing literature suggests that treatment of GCTB with denosumab eliminates tumor giant cells, but tumor stromal cells continue to proliferate and cause histological changes. This may be one of the reasons leading to tumor recurrence or sarcomatous transformation. However, no evidence of its effects on the risk of sarcoma transformation and LR is available as different doses and courses of treatment may have different effects on the histology of GCTB after treatment. How can the dose and course of treatment be determined to reduce histological changes? This remains unclear and is a direction for future research.

## 3. Dose and Duration of Therapy with Denosumab Affect the Blood Supply of GCTB

For patients requiring surgical treatment, effective reduction of tumor blood supply is beneficial for reducing intraoperative bleeding, for better assessment of tumor boundaries, and for surgical resection. Can therapy with denosumab decrease the blood supply to the GCTB? How many doses and what duration of treatment should be utilized for this purpose?

The expression of RANKL, RANK, SATB2, RUNX2, and tumor proliferation and angiogenesis were detected via immunohistochemistry. Girolami I. et al. [33] reported that denosumab exerted antiangiogenic activity in GCTB, which may be mediated through a RANKL-dependent pathway. Some studies [50,51] have also evaluated the effect of denosumab on the blood supply to GCTB using imaging or surgery. However, the pathophysiology of this selective suppression of tumor vascularity and why vascularity is not suppressed elsewhere by denosumab remain unclear. Further research is needed to determine if this mechanism is the same as denosumab-associated osteonecrosis of the jaw (ONJ).

A study included 18 patients with GCTB who received 120 mg of denosumab every month before surgery, with additional doses administered on days 8 and 15 of treatment for a mean duration of 5 months of pre-operative treatment (median, 3 months; range 3–12 months). After 12 weeks of treatment, computer tomography (CT) examination indicated changes in blood supply to sacral or pelvic tumors which were more obvious than those to limb tumors. These changes resulted in a clearer margin of the lesions after treatment, which was helpful for tumor resection [50]. Lim, C. Y. et al. [51] reported that PATD for 68 patients, who received one to four doses with a mean follow-up time of 47.7 months, showed reduced intraoperative bleeding, shortened surgery time, and prevention of early LR. Recurrence-free survival was significantly higher in patients receiving PATD than in those not treated with denosumab in the first 2 years after surgery. However, no significant difference was found following 3 years between the two groups.

A study that assessed tumor vascularization by calculating the CT enhancement rate confirmed that PATD reduced tumor vascularization. This effect was most pronounced early in the treatment course and gradually decreased over time. Therefore, to reduce intraoperative bleeding, PATD is not recommended for more than 3 months before surgery. This information provides a basis for practicians to perform short-term PATD [27]. CT, magnetic resonance imaging (MRI), and positron emission tomography (PET) are useful for evaluating tumor response [52,53] and blood supply. Based on the inverse Choi density/size (ICDS) criteria assessed using CT or MRI images before and after treatment with denosumab, the majority of patients with GCTB have a significant tumor response [54]. Furthermore, PET helps assess early tumor response [55]. Engellau, J. et al. [56] suggested that the improved PET scan criteria and ICDS criteria conferred an improvement in most patients with GCTB, indicating a significantly higher benefit rate compared with the improved Response Evaluation Criteria in Solid Tumors version 1.1 (RECIST) assessment. It seems that the changes in tumor blood supply can be evaluated early using this system. Early identification of the decreased blood supply to the tumor can help surgeons make early decisions about the timing of surgery.

Existing literature shows that PATD can help reduce the blood supply to the tumor, reduce the stage of the tumor, and facilitate surgical resection. Denosumab should not be used more than 3 months before surgery. The information provided here can offer a basis for reducing the dose and duration of PATD, preventing the administration of cumulative doses, and decreasing the incidence of complications. However, the effects of different doses and courses of PDAT on LR remain largely unclear. The effect of denosumab on tumor blood supply needs to be supported by multi-center, large-sample, and prospective studies. This should be demonstrated in future studies.

## 4. Dose and Duration of Denosumab Therapy for GCTB Treatment

Currently, the indications for denosumab related to GCTB mainly include patients with radiologically confirmed recurrent or deemed unresectable GCTB and patients who would have severe complications after resection. PATD provides obvious clinical benefits for patients: tumor size is reduced, surgery is simplified, and joints might be preserved despite the periarticular location of some tumors [57]. Numerous controversies remain regarding the dose and duration of denosumab therapy for different indications [58,59,60].

### 4.1. Dose and Course of Denosumab for Advanced or Inoperable GCTB

For advanced or inoperable GCTB, the usual dose utilized is 120 mg monthly administered by subcutaneous injection, followed by an additional dose on days 8 and 15 after the first dose. Calcium and vitamin D should be concomitantly given for at least 6 months [61]. Several multicenter phase II clinical trials for advanced or inoperable GCTB have been completed and show a definite short-term efficacy [62,63]. However, the appropriate timing of discontinuation of denosumab is subject to various controversies. In most cases, therapy is stopped after serious complications occur, or the disease is controlled. New sarcoma or LR after drug withdrawal is often reported, which warrants serious attention. The incidence of new sarcomas is 1–6% [62,63,64,65]. Chawla, S. et al. [65] reported a study involving 262 patients with unresectable giant cell tumors of bone and a follow-up of up to 5 years, with a local recurrence rate of 11%. A summary of the LR and sarcomatosis rate of denosumab treatment for unresectable GCTB is shown in Table 1.

For patients with unsalvageable or recurrent GCTB or who are likely to have severe complications post-surgery, administration of standard-dose of denosumab provides long-term disease control, with a favorable overall risk-benefit ratio [65]. A phase 2 study (ClinicalTrials.gov: NCT00680992) divided 132 patients with confirmed GCTB of the spine including the sacrum into three cohorts: patients with GCTB that could not be salvaged by surgery (cohort 1), those with planned surgery anticipated to cause severe morbidity (cohort 2), or those with unresectable or recurrent GCTB (no lead-in dosing required) (cohort 3), all with standard dosing regimens of denosumab. The probability of disease progression or relapse in cohort 1 was 3% (95% confidence interval (CI), 0.0–6.2) in the first year and 7.4% (95% CI, 2.1–12.7) in the third and fifth years. Overall, 83% of the patients (all cohorts) reported clinical benefits, including pain reduction, improved mobility, improved function, symptom control, and overall clinical impression compared with the baseline [66]. In a multicenter retrospective analysis, 89 of 138 patients with locally advanced, unresectable, or metastatic GCTB were treated with denosumab at the standard dose for a median of 8 months, and 98% of these patients achieved clinical benefit from treatment. Denosumab has excellent efficacy and tolerability in short-term courses of therapy for patients with unresectable or metastatic disease, but the exact duration of the treatment course remains uncertain [67].

The effect of the standard dose is clear in patients with unresectable GCTB. However, owing to the long course of treatment, serious complications such as mandibular osteonecrosis, atypical femoral fracture, and electrolyte disturbance are prone to occur [68]. Such severe complications frequently require discontinuation of denosumab, which may cause LR of the tumor. In a retrospective analysis of 37 patients with GCTB, 38% showed increased dosing intervals (*n* = 14). The most common dosing interval was 12 weeks (*n* = 8). There were no differences in bone-related complications with prolonged dosing intervals compared with standard-dose denosumab, but 5-year progression-free survival was better (*p* = 0.036). Increasing the dosing interval of denosumab in GCTB provided similar tumor control and reduced bone toxicity events [69]. A comprehensive evaluation is needed to determine when denosumab should be discontinued [70]. Therefore, denosumab is an alternative treatment for inoperable patients. As long-term follow-up data are lacking [71,72], the impact of the dose and duration of denosumab administration on treatment efficacy and incidence of complications should be further studied to achieve better long-term disease control [70,73].

The short-term efficacy of the standard dosing regimen is established in patients with GCTB who cannot be surgically treated. Prolonged use may lead to related serious complications. Discontinuation of the drug is prone to LR. However, prolonged denosumab dosing interval provides similar efficacy with fewer bone-related complications. Studies with larger samples are needed to better determine the optimal interval for denosumab dosing and the impact on efficacy, toxicity, and associated medical costs [69,74].

### 4.2. Dose and Duration of PATD for GCTB

Currently, PATD mainly aims to increase bone mass and reduce tumor size. Thus, unresectable GCTB can be downstaged and surgically treated, and the joint can be preserved after simplified surgery. There are many debates regarding the indications and risks of PATD, especially on whether it increases the risk of LR [75,76]. In most studies, the duration of PATD was 6 months or longer, and the LR rate of patients (42–64%) who underwent curettage with denosumab before the operation was significantly higher than that of patients (11–21%) who underwent curettage alone [77,78,79,80,81]. However, at a mean preoperative time of 3 months, different results were reported. Chinder et al. [82] reported that the LR rate of curettage after preoperative medication (43%) was more than that of curettage alone (18.5%). While, Deventer, N et al. [83] reported that the LR rate of curettage after preoperative medication (28.6%) was lower than that of curettage alone (42.2%) (Table 2).

Patients with LR involving soft tissues, treated with denosumab for six months before surgery, have eggshell-like mineralization of the lesions, which is conducive to lesion resection, reducing the risk of tumor cell extravasation. There was no sign of recurrence four years after the second surgery [84]. In a prospective nonrandomized controlled trial investigating the role of denosumab on preserving joint function, 20 patients received PATD for 6–11 months, while 18 patients were treated with surgery. The median follow-up time was 30 months. The results indicated that PATD provides a favorable and consistent clinical and radiographic response, facilitating less aggressive surgical treatment and joint preservation. However, the LR rate of GCTB after resection does not appear to be affected by denosumab and remains a matter of concern [85,86]. Surgery is difficult for central sites, such as the spine and sacrum, and most tumors in these locations may be difficult to remove. However, in these patients, PATD determines a decrease in the tumor size and blood supply, thus facilitating the surgical excision of the tumor and effective adjuvant treatment [87,88,89]. Another retrospective analysis, which included 58 patients with distal radius GCT and had a follow-up period of 95.3 ± 100.6 months (21–321 months), found soft tissue invasion and tumor size were independent risk factors for LR. PATD was not identified as a risk factor for LR [90]. This may be related to the different sites of tumorigenesis.

For periarticular GCTB, PATD combined with local curettage can be used to achieve joint preservation. Some studies [81] have suggested that a short course of PATD can facilitate surgery, making curettage or resection technically easier, although it is unlikely to improve local control. Extreme LR risk deserves attention. Using denosumab at a standard dose for 6 months before surgery is associated with the LR rate of 15% at 30 months of follow-up and up to 44% at 57 months. One patient developed secondary osteosarcoma and another developed benign GCT pulmonary metastases. Therefore, PATD poses a significant risk (44%) of long-term recurrence and should be cautiously considered before joint preservation surgery [77]. A meta-analysis involving 672 patients with GCTB showed that PATD followed by curettage alone was associated with a higher LR risk than controls. Therefore, they suggested that PATD could increase the LR risk of GCTB when associated with the scraping of the tumor and should be used with caution in this clinical situation [91].

Furthermore, some studies have reported that late recurrence of GCTB is an independent risk factor for malignant transformation, which may be catastrophic, especially for young people; therefore, it is recommended to prudently use PATD with curettage in the treatment of periarticular GCTB [92]. The duration of preoperative adjuvant medication was 6 months. In recent years, there have been reports of short-course pre-operative adjuvant therapy (≤3 months). A retrospective study comparing preoperative ultra-short courses (≤3 doses) and conventional courses (>3 doses) of PATD for sacral GCTB suggested that the former could elicit radiological and histological responses like conventional courses. A smaller degree of fibrosis and ossification facilitates nerve-sparing surgery and contributes to achieving satisfactory local control and functional status while reducing LR risk [26]. For joint-salvage surgery, a short course of PATD (≤three doses) had similar clinical scores, histological and radiological responses, or relapse-free survival as longer courses of therapy (>three doses). Moreover, fewer preoperative doses can reduce complications and treatment costs. However, denosumab should still be used with caution before scraping for GCTB and only when the benefit of joint repair outweighs the likelihood of LR [93]. In patients with unresectable or recurrent GCTB, short-term PATD (≤six doses) improves clinical symptoms, decreases tumor size, and increases tumor density. This simplifies the tumor resection procedure, thereby reducing the LR risk. In the case of curettage, denosumab-induced changes are mixed, and short-term (≤six doses) use may be more appropriate. Although the six-dose regimen is considered safe, its long-term safety remains unknown [94] (Table 3).

Therefore, reducing the dose and duration of PATD can increase bone mass, reduce tumor stage, and reduce the LR rate, achieving the same therapeutic effect as traditional courses while reducing the risk of complications. However, given the limited number of patients, potentially clinically meaningful differences may have been overlooked. Therefore, larger multicenter prospective trials are needed to confirm this, which is the direction of the additional research needed in the future. In particular, two situations should be investigated: on the one hand, to preserve the joint, sufficient bone mass must be restored under the articular surface; on the other hand, for cases where the joint cannot be preserved, the lesion should be mineralized, which is conducive to complete surgical resection. Determining the length of treatment leading to these results requires further research [95].

### 4.3. Dose and Duration of Denosumab Associated with Complications

The most common adverse events of grade 3 or more in 532 patients with GCTB treated with denosumab were hypophosphatemia, ONJ, limb pain, and anemia [65]. ONJ is an uncommon but serious treatment-limiting adverse event [96,97]. The use of denosumab was suspended once ONJ is identified, and it is still a matter of debate whether denosumab can be used again after ONJ was treated [98]. Twenty-nine patients with advanced, inoperable GCTB were treated with denosumab on a standard dosing regimen with a mean follow-up of 70 months (range 1–125 months), and four of these patients (13.8%) experienced medication-related ONJ during treatment [99]. A 15-year-old boy with an unresectable sacral GCTB was treated with the standard dose for 3.6 years and developed ONJ after a cumulative dose of 5520 mg. In addition, complications, such as atypical femoral fractures may also occur [68]. Most serious complications occur in patients with unresectable tumors, owing to a long duration of treatment and high cumulative doses [100,101]. However, for PATD, the greatest risk remains LR, with the possibility of malignant transformation, which can lead to serious consequences. Therefore, further studies on the dose and duration of denosumab in the treatment of GCTB are urgently needed to maximize its therapeutic effect and minimize the risk of complications and recurrence.

## 5. Denosumab Combination Therapy for GCTB

Recent evidence highlights greater efficacy for denosumab administration in combination with a TKI inhibitor like lenvatinib, since angiogenesis is a major hallmark of tumorigenesis, and the involvement of VEGFR (target of lenvatinib) is related to RANKL-induced osteoclastogenesis [102]. This is also of particular relevance to the reduction of blood supply to manage intraoperative bleeding. Indeed, the efficacy of lenvatinib is currently under evaluation in combination with pembrolizumab or ifosfamide/etoposide for the treatment of some bone lesions including osteosarcoma and chondrosarcoma (NCT04784247) [103]. Denosumab combined with everolimus, an mTOR inhibitor, is more effective than denosumab alone for osteoclast differentiation, significantly decreasing bone resorption and exerting bone protective effects [104,105]. Moreover, the secretion of CSF-1 in osteoclast precursors significantly increases after treatment with denosumab. This suggests that 5H4 (an antibody directed against CSF-1) combined with denosumab can increase antitumor efficacy and reduce bone-related events, and warrants investigations in the future [106]. Taken together, there are only a few reports on the treatment of GCTB with denosumab combined with other drugs; however, based on the available evidence, it is speculated that the combination therapy can improve the antitumor effect, and reduce the complications and LR in GCTB. Collectively, it is a promising research direction for future studies.

## 6. Conclusions

Denosumab is an anti-RANKL drug approved by the FDA for the treatment of advanced or unresectable GCTB. Denosumab treatment of GCTB eliminates tumor giant cells; however, tumor stromal cells continue to proliferate, which can cause histological changes in GCTB. The presence of H3A gene mutations in tumor cells after denosumab treatment is helpful in the diagnosis of GCTB. Most studies have shown that denosumab is an effective treatment for unresectable GCTB with definite short-term efficacy. However, long-term therapy with denosumab was discontinued because a long duration of administration and high cumulative doses are risk factors for LR. Therefore, reducing the dose and extending the time interval between denosumab administrations may achieve tumor control in patients who require long-term therapy. This will be a direction for future research.

For patients who will undergo surgical resection, PATD reduces the tumor’s blood supply. Maximum effectiveness was achieved after 3 months of treatment, and lesion mineralization was favorable for tumor resection. Ultra-short-term PATD not only achieves the therapeutic effect of a conventional course of treatment but also reduces the risk of LR. Furthermore, PATD can reduce tumor size, resulting in the downstage of the tumor and facilitating surgical excision of the tumor. For patients needing joint preservation, therapy is recommended only if the benefits of joint preservation outweigh the LR risk. Denosumab should be used with caution in patients who will undergo curettage. 

In conclusion, the dose and duration of denosumab therapy for GCTB need further research. The purpose herein will contribute not only to maximizing the therapeutic effect but also minimizing the risk of complications and LR, bringing greater benefits to these patients.

## Figures and Tables

**Table 1 cancers-14-05758-t001:** Multicenter phase 2 clinical studies for giant cell tumor of bone.

Author	Year	Case	Age (Years)	Couse (Months)	Follow-Up (Months)	Sarcoma (%)	ONJ (%)	LR (%)	Hypo (%)	Registration Number
Chawla, S. et al. [65]	2019	532	33 (25–45)	32.3 (16.3–59.9)	58.1 (34.0–74.4)	1% (4)	3% (17)	11% (28/262)	5% (24)	NCT00680992
Ueda, T. et al. [62]	2015	17	30 (18–66)	≥6	13.1 (8.9–17.9)	6% (1)	0	NR	6% (1)	JapicCTI-111665
Chawla, S. et al. [63]	2013	282	32 (24–45)	13 (7–20)	13 (5.8–21)	1% (3)	1% (3)	NR	5% (15)	NCT00680992
Thomas, D. et al. [64]	2010	37	30 (19–63)	≥6	13	2.7% (1)	no	NR	NR	NCT00396279

Note: LR: local recurrence; ONJ: osteonecrosis of the jaw; Hypo: hypocalcemia; NR: not reported.

**Table 2 cancers-14-05758-t002:** Summary of preoperative denosumab combined with curettage in the treatment of operable GCTB.

Author	Year	Case	Age (Years)	Pre-D (Months)	Follow-Up (Months)	D + C LR%	Time from Post-Op to LR (Months)	C LR%	Time from Post-Op to LR (Months)
Deventer, N et al. [83]	2022	115	33.9 (10–77)	3	65.6(24–404)	28.6% (4/14)	26 (3–86)	42.2% (38/90)	20.1 (2–117)
Asano N et al. [76]	2022	234	34 (28–46)	5 (4–10)	24 (17–30)	64.7%	18	20.1%	18
Perrin, D. L et al. [77]	2021	25	33.8 (18–67)	6	57 (13–88)	44%	NR	NR	NR
Chinder et al. [82]	2019	123	29.6 ± 9.8	3 (1–7)	35 (8–55)	43% (18/42)	12.9 ± 6.5	18.5% (15/81)	14.3 ± 4.9
Puri A et al. [81]	2019	44	27 (13–47)	5 (2–7)	34 (24–48)	44% (11/25)	16 (8–25)	NR	NR
Agarwal MG et al. [78]	2018	52	32 (17–67)	6 (3–17)	27 (12–42)	44% (11/25)	NR	21% (7/34)	NR
Errani et al. [79]	2018	247	29.2 (23–38.5)	≥6	85.6 (54.3–125.1)	60% (15/25)	15 (11–24)	16% (36/222)	15 (9–43)
Scoccianti et al. [80]	2018	21	30 (17–66)	7 (4–7)	23 (7–54)	42% (5/12)	26 (7–54)	11% (1/9)	14

Note: Pre-D: preoperative denosumab; D + C LR%: local recurrence rate of preoperative denosumab combined with curettage; C LR%: local recurrence rate of only curettage; NR: not reported.

**Table 3 cancers-14-05758-t003:** Summary of short-term PDAT for surgically GCTB.

Author	Year	Group	Case	Age (Year)	Doses	Pre-D (Doses)	Follow-Up (Month)	LR%	Level
Liang, H. et al. [26]	2022	short course	41	31.3 ± 10.6	a	≤3	30.3 ± 14.6	8.8	III
		conventional	25	31.7 ± 10.7	b	>3	28.1 ± 15.8	20.8	
Hindiskere, S. et al. [93]	2020	short course	48	30 ± 6.1	a	≤3	37 ± 11.4	27	III
		conventional	36	30 ± 6.3	b	>3	64 ± 15.7	36	
Zhang, R. Z. et al. [94]	2019	short course	11	38.1	c	<6	30 (13–45)	27.2	IV

Note: Pre-D preoperative denosumab; LR%: local recurrence rate.; a: 120 mg every 2 weeks for 1 or 2 months; b: 120 mg monthly for 3–6 months with additional doses on days 8 and 15; c: 120 mg denosumab monthly, with additional doses on days 8 and 15 of the first cycle, six doses in total.

## Data Availability

Not applicable.

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
