# Peer review of "Progress on Denosumab Use in Giant Cell Tumor of Bone: Dose and Duration of Therapy"

_cancers, 2022, doi:10.3390/cancers14235758_

Round 1

Reviewer 1 Report

Thank you for giving me an opportunity to review your manuscript. This is an excellent review paper focusing on denosumab therapy for giant cell tumor of bone. I agree with the contents. There was nothing in particular to correct.

Line19 and 74: This statements is inappropriate because the underlying literature [26] reports no significant difference in local recurrence rates between 3 or more and less than 3 preoperative denosumab doses.

Author Response

Dear reviewer,
Thanks for the constructive critiques as well as the opportunity to improve our manuscript. We have carefully considered all comments and suggestions and revised our manuscript as suggested. The detailed response to the comments and suggestions are listed below point-by-point.

Thanks for your suggestion. The manuscript has been revised accordingly.

Lines 19 and 32: Short-term administration helps attain satisfactory local control and functionality.

Lines 74: Conversely, a short pre-operative course of denosumab (≤3 doses) may also achieve satisfactory local control and functionality in GCTB.

Reviewer 2 Report

The authors provide a comprehensive review of literature about the use of denosumab in the therapy of giant cell tumor of bone. In particular, they focused on the effect of denosumab on histology and on the effect of dose and duration of therapy in relation to blood supply, to the setting (as pre-operative adjuvant therapy or in advanced/inoperable GCTB) and to complications. 

The manuscript is of interest since it deals with a disease which is tipically considered a benign neoplasm due to its indolent behavior, but still it is locally aggressive and, importantly, it often causes severe bone resorption as a result of RANKL signaling promotion of multinuclear osteoclast generation.

The paper is well written and organized.

However, it is strongly recommended that the authors add a paragraph in which they review and discuss the available literature on the combination therapy for bone lesions, since several recent evidences exist about the effective role of TKI and mTOR inhibitors combined with anti-RANKL agents.

For example, recent evidences highlighted an even greater efficacy when denosumab is administered in combination with a TKI inihibitor such as lenvatinib, since angiogenesis is one of the major hallmarks of tumorigenesis, and the involvement of VEGFR (target of lenvatinib) has been related to RANKL-induced osteoclastogenesis. This is also of particular relevance in relation to reduction of blood supply to manage intraoperative bleeding, as mentioned by the authos in section 3. Indeed, efficacy of lenvatinib is currently under evaluation in the treatment of some bone lesions such as osteosarcoma and chondrosarcoma (NCT04784247) in combination with pembrolizumab or ifosfamide/etoposide.

In this regard, please add and discuss the following references:

- Gaspar, N. et al. Lenvatinib with etoposide plus ifosfamide in patients with refractory or relapsed osteosarcoma (ITCC-050): A multicentre, open-label, multicohort, phase 1/2 study. Lancet Oncol. 2021, 22, 1312–1321

- De Vita, A et al: “A Rationale for the Activity of Bone Target Therapy and Tyrosine Kinase Inhibitor Combination in Giant Cell Tumor of Bone and Desmoplastic Fibroma: Translational Evidences.” Biomedicines vol. 10,2 372. 3 Feb. 2022, doi:10.3390/biomedicines10020372

In addition, also combination of denosumab with everolimus, an mTOR inhibitor, has proved more effective than deno alone in osteoclast differentiation, significantly decreasing bone resorption and exerting a bone protective effect.

In this regard, please add and discuss the following references:

- Jeong, H. et al: Final results of the randomized phase 2 LEO trial and bone protective effects of everolimus for premenopausal hormone receptor-positive, HER2-negative metastatic breast cancer. International journal of cancer, 10.1002/ijc.33613.

- Mercatali L et al: The Effect of Everolimus in an In Vitro Model of Triple Negative Breast Cancer and Osteoclasts. Int J Mol Sci. 2016 Nov 1;17(11):1827. doi: 10.3390/ijms17111827. PMID: 27809291; PMCID: PMC5133828.

Moreover, please also include a brief discussion oh the promising role of chemotherapy with CSF-1 based on primary and metastatic bone lesions, as previously reported:

- Liverani C, Mercatali L, Spadazzi C, et al.CSF-1 blockade impairs breast cancer osteoclastogenic potential in co-culture systems. Bone. 2014;66:214-222. doi:10.1016/j.bone.2014.06.017

Author Response

Dear reviewer,
Thanks for the constructive critiques as well as the opportunity to improve our manuscript. We have carefully considered all comments and suggestions and revised our manuscript as suggested. The detailed response to the comments and suggestions are listed below point-by-point.

Thank you for providing these references; we think it is indeed a promising research direction. We will pay further attention to this area in the future. Relevant content and references have been added to the manuscript as suggested.

Line 348-366

  1. Denosumab combination therapy for GCTB

Recent evidence highlights greater efficacy for denosumab administration in combination with a TKI inhibitor like lenvatinib, since angiogenesis is a major hall-mark of tumorigenesis, and the involvement of VEGFR (target of lenvatinib) is related to RANKL-induced osteoclastogenesis[102]. This is also of particular relevance to the reduction of blood supply to manage intraoperative bleeding. Indeed, the efficacy of lenvatinib is currently under evaluation in combination with pembrolizumab or ifosfamide/etoposide for the treatment of some bone lesions including osteosarcoma and chondrosarcoma (NCT04784247)[103]. Denosumab combined with everolimus, an mTOR inhibitor, is more effective than denosumab alone for osteoclast differentiation, significantly decreasing bone resorption and exerting bone protective effects[104, 105]. Moreover, the secretion of CSF-1 in osteoclast precursors increases significantly after treatment with denosumab. This suggests that 5H4 (an antibody directed against CSF-1) combined with denosumab can increase antitumor efficacy and reduce bone-related events, and warrants investigations in future[106]. Taken together, there are only a few reports on the treatment of GCTB with denosumab combined with other drugs but based on the available evidence, it is speculated that the combination therapy can improve the antitumor effect, and reduce the complications and LR in GCTB. Collectively, it is a promising research direction for future studies.

Line 641-655

  1. De Vita A, Vanni S, Miserocchi G, Fausti V, Pieri F, Spadazzi C, Cocchi C, Liverani C, Calabrese C, Casadei R, Recine F, Gurrieri L, Bongiovanni A, Ibrahim T, Mercatali L. A Rationale for the Activity of Bone Target Therapy and Tyrosine Kinase Inhibitor Combination in Giant Cell Tumor of Bone and Desmoplastic Fibroma: Translational Evidences. Biomedicines. 2022,10.
  2. Gaspar N, Venkatramani R, Hecker-Nolting S, Melcon SG, Locatelli F, Bautista F, Longhi A, Lervat C, Entz-Werle N, Casanova M, Aerts I, Strauss SJ, Thebaud E, Morland B, Nieto AC, Marec-Berard P, Gambart M, Rossig C, Okpara CE, He C, Dutta L, Campbell-Hewson Q. Lenvatinib with etoposide plus ifosfamide in patients with refractory or relapsed osteosarcoma (ITCC-050): a multicentre, open-label, multicohort, phase 1/2 study. Lancet Oncol. 2021,22:1312-1321.
  3. Jeong H, Jeong JH, Kim JE, Ahn JH, Jung KH, Koh SJ, Cheon J, Sohn J, Kim GM, Lee KS, Sim SH, Park IH, Kim SB. Final results of the randomized phase 2 LEO trial and bone protective effects of everolimus for premenopausal hormone receptor-positive, HER2-negative metastatic breast cancer. Int J Cancer. 2021.
  4. Mercatali L, Spadazzi C, Miserocchi G, Liverani C, De Vita A, Bongiovanni A, Recine F, Amadori D, Ibrahim T. The Effect of Everolimus in an In Vitro Model of Triple Negative Breast Cancer and Osteoclasts. Int J Mol Sci. 2016,17:1827.
  5. Liverani C, Mercatali L, Spadazzi C, La Manna F, De Vita A, Riva N, Calpona S, Ricci M, Bongiovanni A, Gunelli E, Zanoni M, Fabbri F, Zoli W, Amadori D, Ibrahim T. CSF-1 blockade impairs breast cancer osteoclastogenic potential in co-culture systems. Bone. 2014,66:214-222.

Reviewer 3 Report

The English writing needs a bit of clarification but in general is very good. The manuscript is a bit too long and unnecessarily complicated. Some of the references have been used incorrectly. It would be nice if the authors can summarize not only the evidence of selective suppression of tumor vascularity by Denosumab, but the pathophysiology of this suppression and why vascularity isn't suppressed elsewhere by Denosumab (or is it? do the authors think that this is the same mechanism for AVN of the jaws?).

Author Response

Dear reviewer,
Thanks for the constructive critiques as well as the opportunity to improve our manuscript. We have carefully considered all comments and suggestions and revised our manuscript as suggested. The detailed response to the comments and suggestions are listed below point-by-point.

Point 1: The English writing needs a bit of clarification but in general is very good.

Response 1:

Thanks for your suggestion. Accordingly, we have revised it again and asked a native English expert to polish the language.

Point 2: The manuscript is a bit too long and unnecessarily complicated.

Response 2:

Thanks for your suggestion. Accordingly, we have omitted some phrases to reduce wordiness and achieve conciseness.

Point 3: Some of the references have been used incorrectly.

Response 3:

Thanks for your suggestion. We did cite some references incorrectly and have corrected them in the revised manuscript as follows:

Line 59

fracture, and bone metastases associated with solid tumors[10-12]

  1. Clézardin P, Coleman R, Puppo M, Ottewell P, Bonnelye E, Paycha F, Confavreux CB, Holen I. Bone metastasis: Mechanisms, therapies, and biomarkers. Physiological reviews. 2021,101:797-855.

changed to

  1. Fizazi K, Carducci M, Smith M, Damião R, Brown J, Karsh L, Milecki P, Shore N, Rader M, Wang H, Jiang Q, Tadros S, Dansey R, Goessl C. Denosumab versus zoledronic acid for treatment of bone metastases in men with castration-resistant prostate cancer: a randomised, double-blind study. Lancet. 2011,377:813-822.

Line 341

complications occur in patients with unresectable tumors, owing to a long duration of treatment and high cumulative doses[100, 101].

  1. Otto S, Pautke C, Van den Wyngaert T, Niepel D, Schiødt M. Medication-related osteonecrosis of the jaw: Prevention, diagnosis and management in patients with cancer and bone metastases. Cancer Treatment Reviews. 2018,69:177-187.

changed to

  1. Ruggiero SL, Dodson TB, Fantasia J, Goodday R, Aghaloo T, Mehrotra B, O'Ryan F. American Association of Oral and Maxillofacial Surgeons position paper on medication-related osteonecrosis of the jaw--2014 update. J Oral Maxillofac Surg. 2014,72:1938-1956.

Line144

Thus, it reduces blood supply to GCTB[50]

changed to

Some studies have also evaluated the effect of denosumab on the blood supply to GCTB by imaging and/or surgery.

deleted

  1. Lin P, Lin N, Teng W, Wang SD, Pan WB, Huang X, Yan XB, Liu M, Li HY, Li BH, Sun LL, Wang Z, Zhou XZ, Ye ZM. Recurrence of Giant Cell Tumor of the Spine after Resection: A Report of 10 Cases. Orthop Surg. 2018,10:107-114.

Point 4: It would be nice if the authors can summarize not only the evidence of selective suppression of tumor vascularity by Denosumab, but the pathophysiology of this suppression and why vascularity isn't suppressed elsewhere by Denosumab (or is it? do the authors think that this is the same mechanism for AVN of the jaws?).

Response 4:

Thanks for your suggestion. We searched the literature on the anti-angiogenic activity of denosumab and Girolami I et al. for the first time reported that denosumab has antiangiogenic activity in GCTB, which may be mediated through the RANKL-dependent pathway. Relevant content has been added to the revised manuscript. Unfortunately, no more studies on the mechanism of the anti-angiogenic activity of denosumab were found.

Line141-145

IHC was performed to evaluate the expressions of RANKL, RANK, SATB2, and RUNX2, along with tumor proliferation and angiogenesis. Girolami I et al[33] first reported that denosumab exerted antiangiogenic activity in GCTB, which may be mediated through a RANKL-dependent pathway. Some studies have also evaluated the effect of denosumab on the blood supply to GCTB by imaging and/or surgery.

Line 477

  1. Girolami I, Mancini I, Simoni A, Baldi GG, Simi L, Campanacci D, Beltrami G, Scoccianti G, D'Arienzo A, Capanna R, Franchi A. Denosumab treated giant cell tumour of bone: a morphological, immunohistochemical and molecular analysis of a series. J Clin Pathol. 2016,69:240-247.

The mechanism of medication-related AVN of the jaws remains unclear, and several possibilities have been proposed. It is believed that infection and subsequent inflammation play crucial roles. Other proposed mechanisms include the suppression of bone remodeling, angiogenesis, the proliferation of oral mucosal cells, and aberrant immune function.

The RANKL signaling pathway may also mediate necrosis of mandibular osteonecrosis caused by denosumab. Thus, the mechanism of the anti-angiogenic activity of denosumab may be similar to that AVN of the jaws and warrants further investigation.

Reference

[1]Girolami I, et al. Denosumab treated giant cell tumour of bone: a morphological, immunohistochemical and molecular analysis of a series. J Clin Pathol. 2016,69:240-247.

[2]Allegra A, et al. Antiresorptive Agents and Anti-Angiogenesis Drugs in the Development of Osteonecrosis of the Jaw. Tohoku J Exp Med. 2019,248:27-29.

[3]Otto S, et al. Medication-related osteonecrosis of the jaw: Prevention, diagnosis and management in patients with cancer and bone metastases. Cancer Treat Rev. 2018,69:177-187.

[4]Lesclous, P. et al. Bisphosphonate-associated osteonecrosis of the jaw: A key role of inflammation? Bone.2009,45:843–852.

[5]Allen, M. R. & Burr, D. B. Mandible Matrix Necrosis in Beagle Dogs After 3 Years of Daily Oral Bisphosphonate Treatment. J. Oral Maxillofac. Surg.2008,66:987–994.

[6]Gkouveris, I. et al. Vasculature submucosal changes at early stages of osteonecrosis of the jaw (ONJ). Bone.2009,123:234–245.

[7]Soydan, S. S. et al. Effects of alendronate and pamidronate on apoptosis and cell proliferation in cultured primary human gingival fibroblasts. Hum. Exp. Toxicol. 2015,34:1073–1082.

[8]Zhu, W. et al. Zoledronic acid promotes TLR-4-mediated M1 macrophage polarization in bisphosphonate-related osteonecrosis of the jaw. FASEB J.2019, 33:5208–5219.

Round 2

Reviewer 2 Report

The authors addressed all the issues raised in previus revision, therefore the manuscipt improved significantly. 

Author Response

thank you!